# Changes in Physical Activity Levels and Eating Behaviours during the COVID-19 Pandemic: Sociodemographic Analysis in University Students

**DOI:** 10.3390/ijerph19095550

**Published:** 2022-05-03

**Authors:** Maria Ferrara, Elisa Langiano, Lavinia Falese, Pierluigi Diotaiuti, Cristina Cortis, Elisabetta De Vito

**Affiliations:** Observatory on Health Education and Hygiene Laboratory, Department of Human Sciences, Society and Health, University of Cassino and Southern Lazio, 03043 Cassino, Italy; m.ferrara@unicas.it (M.F.); langiano@unicas.it (E.L.); p.diotaiuti@unicas.it (P.D.); c.cortis@unicas.it (C.C.); devito@unicas.it (E.D.V.)

**Keywords:** healthy lifestyle, young adults, COVID-19, physical activity, exercise, sedentary behaviour, diet, eating habits, university students

## Abstract

The COVID-19 pandemic has forced schools and universities to shift their activities online, influencing the adoption of health-related behaviours such as physical activity and healthy dietary habits. The present study investigates the changes in adherence to a healthy diet and regular physical activity in university students in Italy before and during the COVID-19 pandemic and understands the role of sociodemographic variables in creating the changes above. We conducted a repeated cross-sectional survey performing the same sampling strategy at the first data collection (T0) and second data collection (T1) with a combination of convenience and snowball sampling approaches. The sample is composed of a total of 2001 students, 60.2% women and 39.8% men, with an average age of 22.7 (±5.5 SD). At T1, 39.9% of the students reported regular physical activity. During the pandemic, however, many, especially male students, abandoned or reduced physical activity practice (T1 40%), with an increase in social media use (T0 52.1%; T1 90%). A direct association between very low frequency of physical activity and increased sedentary time (r = 0.2, *p* = 0.001) and between change in dietary style and increased Body Mass Index (BMI) value (r = 0.3, *p* = 0.002) was found. The multivariate analysis for the total sample showed that some sociodemographic variables such as gender, age, parents’ level of education, area of study, household type, and perception of one’s body influence eating behaviours and physical activity. Our findings suggest that universities should invest in the protection and promotion of the health of their students with specific awareness programmes, and further research should repeat the survey in the post-lockdown period to investigate the long-term effects on health-related behaviours.

## 1. Introduction

Modifiable risk factors related to unhealthy behaviours and lifestyles, such as tobacco use, unhealthy diet, lack of physical activity (PA), and alcohol abuse, among others, are associated with many chronic conditions and with an onset of non-communicable diseases (NCDs) causing the majority of deaths worldwide, regardless of age, sex or geographic origin [1,2,3]. Around 75% of premature deaths caused by NCDs occur in adults aged 30–69 years, demonstrating that NCDs are not only a problem for older people [3,4,5].

The majority of risky behaviours are indeed established at an early age and are consolidated in adulthood. During adolescence, youth begin to develop habits that will carry over into adulthood with considerable repercussions on their risk for NCDs [4,6]. Therefore, adolescents and young adults represent the most important target for preventive intervention of NCDs. The transition from high school to university is also a critical stage in the development of health-related behavioural habits, and previous studies found that in this phase of their life, university students are prone to adopt unhealthier behaviours [7,8,9].

Compliance with a healthy lifestyle, especially the adherence to a combination of more than one healthy behaviour, is associated with a lower risk of mortality and a reduced risk of NCDs [5,10,11,12,13]. The most effective measure available to tackle NCDs is, therefore, their prevention at all stages of life, which represents a significant public health challenge worldwide.

An unhealthy diet and physical inactivity are among the most critical risk factors for NCDs. It has also been established that the combination of physical activity and healthy nutrition has the best health benefits and modulates health throughout the lifespan [14,15,16].

Physical inactivity represents the fourth most common risk factor for deaths worldwide, responsible for approximately 3.2 million deaths each year. Physically inactivity has been associated with an unfavourable cardiovascular disease risk profile, including obesity, insulin resistance, and high blood pressure [17,18].

To encourage this highly protective health behaviour, in 2018, the World Health Assembly (WHA) approved a new *Global Action Plan on Physical Activity (GAPPA) 2018–2030*^.^ It adopted a new voluntary global target to reduce international levels of physical inactivity in adults and adolescents by 10% by 2025 and 15% by 2030 [19]. Nevertheless, one in four (27.5%) adults and more than three-quarters (81%) of adolescents worldwide do not practice enough PA [20], and this seems to be true also for university students who can exceed 9 h per day of sedentary time [21,22,23] and decrease the practice of physical activity [24,25,26].

In Italy, according to the National Institute of Statistics (ISTAT), in 2019, the percentage of sedentary people was 35.6%, and the practice of physical activities and sport decreased with age [27].

Unhealthy eating behaviours, excess body weight, excessive consumption of energy, saturated fats, trans fats, sugar, and salt, and low consumption of vegetables, fruits, and whole grains are also leading risk factors and significant public health concerns [28]. Food selections and dietary patterns such as skipping meals, under-eating, or over-eating can lead to a decrease in diet quality and increased risk of chronic diseases [29].

Starting from the beginning of 2020, the restrictive measures put in place by governments to limit the spread of the SARS-CoV2 virus worldwide and, more specifically in Italy, strongly affected PA and nutrition behaviours [30]. Changes in food consumption, type of food, number of meals and snacks, for example, and a decrease in time spent in physical activity and changes related to the type of activities were recorded during the pandemic [31].

In response to the rapid increase in the number of COVID-19 cases, the Italian government declared a state of emergency on 31 January 2020 [32] under Legislative Decree 1/2018 and on 9 March, Italy was the first European country to enter a nationwide lockdown, which was initially imposed on some northern Italian region, and within days, extended to throughout the country.

The Italian government, like many European countries, put in place unprecedented non-pharmacological community interventions to control and prevent the spread of the disease throughout the country with restrictions more and more severe: from the obligation for everyone to stay at home, banning mass gatherings and public events, closing schools and universities, retail stores, bars and restaurants, encouraging people to work from home and avoiding going out in public, to the limitation of free movement of people, including sport-related activities, walking and running outdoors, to the closure of almost all work activities [33,34,35,36].

All these measures influenced the perception of risk, the perception of individual self-efficacy, the value attributed to social responsibility and the trust in health authorities and others, and the adoption of certain health-related behaviours [37].

Most of the restrictive measures had limited the participation in physical activity, sport, and exercise with a resulting increase in sedentary behaviours and inactivity levels, which may lead to increased risk for physical and mental health problems [30,37,38,39,40,41].

University students who often use active commuting to reach the university for short and medium distances reduced their daily energy expenditure and increased the time spent sitting to listen to online classes and study from home [38].

To counteract inactivity and sedentary behaviours, experts recommend taking any chance to walk and stand up, do home-based physical activities and exercise, and try to be regular [42]. Playing active video games (AVGs) could also be a valuable strategy to reduce sedentary behaviours when it is not possible to do other physical activities outside of the house [43].

Italian studies on eating habits changes during the COVID-19 lockdown affirmed that the sense of hunger and satiety changed for more than half of the population with less appetite or increased appetite; an increase in the intake of sweets, salty snacks, sweet beverages, and alcohol was reported, as well an increase in the consumption of healthy foods, such as fruits and vegetables, extra virgin olive oil, and legumes [30,40,44,45,46,47]. A previous study conducted in Italy showed that, in university students, healthy food consumption and dietary habits during the COVID-19 pandemic were influenced mostly by the practice of exercise and by mental health, including mood states and self-efficacy [46].

Several studies have been conducted on the PA and nutrition of children and adolescents before and during the COVID-19 pandemic worldwide and also in Italy. Still, few researchers investigated the specific population of university students with data collected before and during the pandemic [38,40].

All these reasons have led us to carry out a study aimed at investigating eating behaviours and physical activity levels in university students in Italy before and during the COVID-19 pandemic and investigating whether these behaviours and any changes are influenced by sociodemographic and individual variables such as lifestyle before the pandemic.

Moreover, the findings of the present survey, by assessing the main modifiable risk factors for NCDs, dietary habits, and PA through the self-reported experience of university students, could be helpful in the development of preventive actions for this specific target population.

## 2. Materials and Methods

### 2.1. Design and Selection of Study Subjects

The sample size was selected at convenience without aprioristic statistical calculations and non-probabilistic random sampling.

We conducted a repeated cross-sectional survey by submitting the same questionnaire in two different periods, before and during the pandemic [48,49]. Students enrolled in bachelor’s or master’s programmes at universities in central Italy were invited to participate in the study through their student representatives and social media networks such as Facebook and WhatsApp platforms (Meta Platforms, Inc, Menlo Park, CA, USA). In order to recruit a large and diverse sample, no particular groups were targeted, and no exclusion criteria were specified; however, questionnaires that were incomplete or completed by students from universities located in a geographical area different from central Italy were excluded from the analysis.

Detailed information on the purpose of the study and the statement on anonymity were clearly described at the beginning of the questionnaire. Authorisation to process sensitive data (General Data Protection Regulation 2016/679) [50] and informed consent were mandatory fields to continue the survey.

The first data collection took place between November 2018 and February 2019 (T0). Students were asked to fill in a questionnaire in a paper format containing information on sociodemographic data and lifestyles (physical activity, eating habits, tobacco smoking, alcohol use and substance abuse, sexual behaviours). At T0, we collected data from 1025 students (35.5% men 64.5% women), with an average age of 22.6 years old (±3.6 SD).

The second data collection took place online, during the COVID-19 pandemic, between November 2020 and February 2021 (T1). At T1, we collected data from 976 students (31.3% men, 68.7% women), with an average age of 21.3 years old (±4.1 SD). The questionnaire was uploaded on the Google Form platform, and the same sampling strategy was performed for the recruitment of T0 students. We used the same questionnaire as in the first data collection, but we decided to exclude questions about behaviours other than physical activity and eating habits and to add some questions about media and leisure time activities during the pandemic and the perceptions of change in PA and eating behaviours.

For the two data collections, the same sampling strategy was used with a combination of convenience and snowball sampling approaches. The availability of data before and during the pandemic and the use of the same survey instrument and two samples with very similar characteristics (socio-demographic, PA and eating habits) justified the sample size and selection and made the subsamples statistically comparable.

### 2.2. Survey Tool

The questionnaires were created ad hoc, in Italian, by the Health Education Observatory of the Hygiene Laboratory of the Department of Human Sciences, Society and Health of the University of Cassino and Southern Lazio. They included adapted questions on health behaviours from the Health Behaviour in School-aged Children (HBSC) survey [51]. The questionnaire was initially submitted to school-aged students and university students. Only the university students were included in the analysis for the present study.

The first version of the questionnaire consisted of 125 items, divided into six sections. The first section (I) gathered sociodemographic and family-related data (gender, age, area of residence, parents’ level of education and occupation, family environment, etc.). It used categories defined by ISTAT [52]. Section two (II) included information about the use of drugs, followed by details regarding tobacco smoking habits and the consumption of alcoholic beverages (III and IV sections). Reproductive health and sexual behaviours were the main topics of the fifth (V) section, while section VI focused on physical activity and eating habits.

In this last section, students were asked to indicate if they performed any PA (yes/no), the frequency (days per week), and the type of PA and sport eventually practised. According to the yes/no answer about the PA practice, we created two categories, sedentary and active, and then, for active students, we made three subcategories of frequency, namely very low frequency (a few times a month–less than one time per week), low frequency (1–2 times per week), and medium-high frequency (three times per week or more).

Eating habits questions included the number of meals, the distribution of meals during the day (heavy/light meals), and the motivation to skip meals, if any.

Weight and height data were self-reported and used to calculate the Body Mass Index (BMI) and then to define the status of underweight, average weight, overweight, and obese using the International Obesity Task Force (IOTF) thresholds from Cole et al. (2012) [53]. In addition, one question was added to detect the students’ self-perception of their weight status.

The descriptions of the health behaviours we were investigating were reported in the questionnaire according to the definitions used by the WHO and the international survey on the health, well-being, and behaviour of young people “Health Behaviour in School-aged Children” (HBSC) [54,55].

The second version of the questionnaire consisted of 49 items. Items 1–14 (sociodemographic and family-related questions) corresponded to section I of the previously described questionnaire. Items 15–20 investigated physical activity behaviours (type, frequency, motivation) with the same questions as the first version of the questionnaire. Information about eating habits (same questions as the first version) was requested in items 21–31, while the last part of the questionnaire investigated the use of media and leisure time during the pandemic (items 32–49). Questions about the perception of changes in PA and eating behaviours during the COVID-19 pandemic were added to this version of the questionnaire.

### 2.3. Statistical Analysis

A descriptive univariate analysis was performed to represent the dataset synthetically and to describe the sociodemographic and lifestyle characteristics of the two different samples using a simple frequency distribution. A bivariate analysis was performed to investigate the association between sociodemographic factors (gender, age, education level, and parental occupation) and lifestyle.

Exploratory analyses were used to investigate the distribution of the independent variables. Differences between groups were estimated using the Chi-square test and tests without distribution, and those with a *p*-value < 0.05 were considered significant. The values of Cronbach’s alpha (coefficient of internal consistency) and the Mann–Whitney U test were used to determine the mean differences in the perceived change spent in physical activity of the student respondents in the two periods considered (indicated as T0 = before the pandemic and T1 = pandemic).

The calculation of Body Mass Index (BMI = kg/m^2^) and the classification into underweight, average weight, overweight, and obese was carried out according to Cole’s tables, separately for age and gender in both samples [53].

A simple linear regression model assessed the relationship between the dependent variables (sedentary lifestyle and change in BMI value) and the independent variables (physical activity, healthy eating behaviours).

The adjustment for sociodemographic characteristics (age, gender, area of study, parents’ level of education, perception of one’s body) took place through the coding of the sociodemographic variables that could influence the behaviour of our sample, and consequently, some dummy variables were created, and the possible effects of the changes on the dependent variables (PA and Eating Habits) were evaluated.

Appropriate logistic regression models were built to investigate the association between health behaviours adherence and eventual modification during the pandemic (attainment of recommended PA levels and commitment to a good eating pattern) about certain ascertained risk factors such as age (categorised as less than/equal to 25 years or more than 26 years), gender (male or female), perception of one’s body (positive or negative); type of degree programme, including scientific (engineering, mathematics), humanities (humanities, philosophy, education, social work, exercise science) health (medicine, biology, biotechnology, nursing), business/legal; parental education levels categorised as low (≤elementary school), medium (middle school and high school), and high (≥college); and BMI (classified as usual and overweight).

The dependent variable for PA was built with two specific models: the first assigned a dichotomous YES/NO value, “YES” identifying participants who engaged in physical activity and “NO” those who did not engage in any physical activity (sedentary), the second assigned a value of 1 to those who reported engaging in physical activity with a very low frequency (<of 1 time per week), 2 to those with a low frequency (1–2 times per week), and 3 to those who engaged in physical activity with a medium-high frequency (three times per week or more).

The number of meals per day (1-2-3-4-5 or more) was considered for eating habits.

Risk factors were calculated by adjusting odds ratios (ORs) and 95% confidence intervals (CIs).

Statistical analyses were performed using the EpiInfo 3.5 statistical package; the statistical significance level was set at *p* < 0.05.

## 3. Results

One thousand twenty-five students at T0 and 976 at T1 from central Italy responded entirely to the questionnaire. The values of Cronbach’s alpha (coefficient of internal consistency) for the questionnaire used in the study were 0.74 and 0.78, respectively, for the study carried out at T0 and T1. The values obtained showed a satisfactory level of reliability [56]. The total sample consisted mainly of women (60.2% women vs. 39.8% men), with a mean age comparable between the groups. The sociodemographic characteristics of the two samples (T0 and T1) are shown in Table 1.

Table 2 shows descriptive data about the statistically significant changes in the regular practice of PA, eating habits, BMI, and perception of one’s weight status in the total sample and the two subsamples (T0 and T1).

In this study, 39.9% of the total sample (2001) reported practising physical activity regularly before the pandemic; about 65% of them declared to practise with a frequency of three times a week. When analysing the most active segments of the examined population, some statistically significant differences emerged in sex and age in the two groups. Male students (T0 56. 9% vs. 43.3% of females; *p* = 0.001 vs. T1 60.8% vs. 39.2% of females; *p* = 0.002) and the youngest (T0 61.9% of those aged ≤25 years vs. 38.7% of those aged ≥26 years; *p* = 0.03 vs. T1 55.6% of those aged ≤25 years vs. 37.0% of those aged ≥26 years; *p* = 0.03) reported to practise at least one sport regularly. Analysing the frequency of the type of faculty, it emerges that, in the total sample (T0 + T1), those who declare to do physical activity regularly are mainly students enrolled in degree courses in exercise science and health disciplines (73.1% vs. 33.9% other courses; *p* = 0.03), and who live alone (67.2% vs. 39.6% in a family *p* = 0.04). The sports most practised are swimming, jogging, and gymnastics/fitness, and the motivations are to be healthy (T0 40.7%; T1 38.9%), to get rid of stress (T0 34%; T1 32.8%), and to lose weight (T0 26.9%; T1 21.1%), the latter a motivation declared mainly by the female respondents in both groups (67.6%; *p* = 0.007). With the advent of the pandemic, however, many abandoned this practice (T1 40%), with a relative increase in sedentariness and social media use (T0 52.1%; T1 90%) mainly among male respondents in both groups (61.5% vs. 31.8% female; *p* = 0.002). As expected (T1), the pandemic resulted in less time devoted to physical activity (<of 2 times per week, *p* = 0.001) and consequently more sedentary activity, especially in women. In fact, there was a direct association between very low frequency of physical activity and increased sedentary (r = 0.2, *p* = 0.001) and between change in dietary style and increased BMI value (r = 0.3, *p* = 0.002).

The mean weight and mean height of the total sample were within the normal mean values, with differences between the sexes, respectively Kg69.5 ± 15.1 male vs. Kg61.08 ± 10.6 female (*p* = 0.000) and cm184 ± 7.06 male vs. cm170 ± 5.7 female (*p* = 0.000). Weight assessment according to BMI (calculated from self-reported weight and height), reported underweight (T0 2.9% vs. T1 1.8%), normal weight (T0 85.7% vs. T1 81.6%), overweight (T0 9.9% vs. T1 14.3%) and obese (T0 1.5% vs. T1 2.3%), overall males were overweight and females obese (Table 2). The percentage of overweight students seemed to increase with increasing age, from 1.6% in the group of ≤25 years to 2.3% ≥ 26 years (*p* = 0.03).

The subjective perception of one’s own weight status mirrors the objective data. In fact, the overall sample considers itself to be normal weight (65.4%), but with statistically significant differences between the two sexes in relation to overweight (Table 2) and between the different groups of the sample according to the temporal location of 64.4% T0 vs. 35.4% T1 (*p* = 0.004).

As an indicator of eating behaviour, we used the number of meals during the day. In the total sample, the frequency of meals is distributed as follows: 11.1% indicated having five meals, 59.9% had three meals, and 30% had two meals. These percentages vary in the two subgroups (T0 respectively 10.3%, 58.7%, and 31% versus T1 31.3%, 59%, and 7%). Breakfast is regularly eaten by more than 50% of young people before leaving home; this healthy habit decreases with increasing age (from 38.3% of the group ≤25 years to 23.2% of the group ≥26 years *p* = 0.02) and changes according to gender. Women seem to eat breakfast more rarely (Table 2), especially among respondents with working mothers (*p* = 0.001). This difference is more pronounced in the group ≤25 years (41.2%), while it is smaller in the group ≥26 years (29.1%) (*p* = 0.02). The morning snack is consumed by 31% of women and 18.1% of men. The majority of students consume the two main meals of the day, mainly at home with their families. Thirty-nine percent claim to have followed a diet at least once, and most women report this (Table 2). A total of 15.8% of young people at T0 report being on a diet at the time of the survey, which decreases by about five percentage points during the block (T1). Furthermore, the adoption of dieting increases with age and is predominantly practised by women (33.2%) compared to men (15.9%).

After adjustment for sociodemographic characteristics, the logistic regression models show that of the total sample participants being male, being older than 26 years, living in households with both parents with a low level of education, and having a positive body perception are risk factors for adopting a sedentary lifestyle, risk factors confirmed by applying the same regression models to the two subsamples compared at T0 and T1 (Table 3).

In contrast, being younger than 25 years, being female, and living alone are significant predictors of an increased risk of unhealthy eating behaviours in the total number of students who participated in our survey. Again, the same regression models applied at T0 and T1 confirmed the sociodemographic factors are taken into account, except “living alone”, which was not guaranteed to be a risk factor at T1, probably because the lockdown forced most of the students to return to their families. (Table 4).

At T1, we registered a high percentage of physical activities drop out and a change in the way of practising and the type of activity. Students declared to practise mainly those activities allowed by the COVID-19 restriction: outdoor and home fitness. On the other hand, there was a considerable change in the way they performed the activity for those who decided and were able to continue with their sport discipline. In the students analysed at T1, the increase in sedentary time was obviously due to the lockdown period that pushed them to spend more hours in front of the TV and the computer.

## 4. Discussion

Our study aimed at investigating the eating and physical activity behaviours of university students in Italy before and during the COVID-19 pandemic trying to understand whether the extent of change, if any, may depend on specific individual and environmental factors.

The results of our study show changes in physical activity and eating behaviours before and after the pandemic.

Before the pandemic, about 40% of the students were physically active, especially male students and younger students. A decrease in the time spent in physical activities, a change in the type and level of PA, and an increase in sitting time and sedentary behaviours affected both men and women students, but men’s PA behaviours seemed to be more affected by the isolation and quarantine. In our study, being male, older than 26 years old, and living in a household with parents with a low level of education were considered risk factors that led to being less active during the pandemic.

We also found differences in eating behaviours before and during the pandemic with an increase in the number of meals at T1.

Finally, we also found a significant increase in the percentage of overweight and obese students at T1.

The adoption of improper lifestyles in youth could be due to the fact that they tend to underestimate the probability of the negative consequences since they do not think that such events can happen to them [57,58,59]. A recent study found that university students were inclined to lead unhealthy lifestyles, and more specifically, their eating behaviours most of the time did not fulfil the recommendations [9].

According to previous studies, the main external barriers to PA practice in university students are the lack of time due to busy lesson schedules and parents’ pressure on academic performance. Internal barriers such as lack of energy and motivation could also affect the low rates of PA at this age [60].

Concerning eating habits, we found that the majority of the student in the total sample ate three meals per day. The impact of meal frequency on overweight and obesity in children and adults has been previously investigated, but the findings are far from being heterogeneous [61].

Previous studies on weight gain in university students showed that, compared to the same age individuals not attending colleges or universities, students have a higher probability of gaining weight and of being overweight, especially in the first year of university, primarily due to insufficient physical activity, poor diet, and stress that encourages bad habits such added snacks or skipping breakfast [62,63,64]. Although many university students know about balanced diets, increases in sugar, fat, and sodium intake and low consumption of fruits and vegetables appear to be common in this specific population, especially due to improper cooking and eating behaviours [65,66].

The BMI of our sample at T0 was very similar to the one described by Teleman et al. in 2015 for an example of Italian university students. The majority of students were of average weight, and male students were reported to be more overweight than females [21,22].

Both samples (at T0 and T1) mainly reported activities such as swimming, jogging, and fitness, and these findings are in line with the data related to the Italian adult population [67]. Motivations to practise sport and PA for our sample are mostly to be healthy, get rid of stress, and lose weight. Appearance, weight, and stress management are the motivations for exercise practice more reported by young adults and college students [68,69].

As per other recent studies on the impact of the COVID-19 pandemic on university students worldwide [70,71,72,73,74,75], our data showed changes related to PA, eating habits, BMI, and body image perception during the pandemic. More specifically, the descriptive analysis showed that the levels of PA decreased by 40% at T1.

These results confirm the findings of other studies on lifestyles of university students during the COVID-19 pandemic that recorded a decrease in the time spent in physical activities, a change in the type and level of PA, and an increase in sitting time and sedentary behaviours, with some differences depending on several sociodemographic and individual variables [38,44,70,76,77,78,79,80]. The changes in the way of practising physical activities, exercise, and sport during the pandemic were mostly due to the restrictive measure adopted by governments. Outdoor activities and home fitness have replaced many activities that were prohibited. On the other hand, there was a considerable change in the way they performed the activity for those who decided and were able to continue with their sporting discipline due to the unavailability of specific equipment and facilities [44]. However, some research highlighted that, despite the decreases in time spent in physical activity and the changes related to the type of activity, those who were achieving recommended levels of PA before the pandemic would appear to continue to reach them even during the lockdown, especially if younger than 22 years old, female, previously active, and with at least one graduate parent [78,79,81].

According to Brancaccio et al. (2021), the male population was more affected by isolation and quarantine, reporting more unfavourable behavioural changes during the COVID-19 pandemic [81].

We found that women were less active than men both before and during the pandemic, but men’s PA behaviours seemed to be more affected by the isolation and quarantine. The difference in the adherence to PA between sexes confirms data from the Italian National Institute of Statistics [27] and data from other international studies [8]. We found that being male, older than 26 years old, and living in a household with parents with a low level of education were considered risk factors to be less active during the pandemic in our sample. Education was considered an indicator of socioeconomic position, and several studies before the pandemic showed evidence of a positive association with PA, especially during adolescence [82]. On the other hand, a longitudinal study analysing adults’ PA before, during, and after COVID-19 restrictions did not find any significant association between socioeconomic status and changes in PA [83]. Moreover, women, as also demonstrated in another study on the Italian adult population, showed a lower tendency to reduce physical activity levels during the lockdown, revealing greater resilience than men [84].

In our study, eating behaviours also changed during the pandemic. More specifically, the students increased the number of meals at T1. The percentage of students eating five meals per day increased from 10.3% to 31.3%. These data align with the results of recent studies on the impact of the pandemic on dietary habits in various population groups that found that participants increased their meal number and frequency during quarantine [85].

A recent study about the effects of COVID-19 home confinement on eating behaviour, number of meals and snacks, food consumption, and type of food were recorded to have unhealthier patterns than before the pandemic in an international survey of adults [31].

We found that also BMI changed during the pandemic. According to the reported data on weight and height, the BMI calculated showed a significant increase in the percentage of overweight and obesity at T1. As discussed in other studies on the impact of COVID-19 on university students’ BMI, it seemed that the changes in food consumption and physical activity negatively affected the students’ BMI [86,87].

The regression analysis on eating habits for the total sample showed that female students younger than 25 years old and living alone have a higher risk of unhealthy eating behaviours. Our findings agree with the results of a study about dietary habits of university students in Italy that reported more difficulties in adopting a healthy diet in students living alone than the ones living with their families [88]. On the other hand, the interactions between sex and eating habits were not significant in a previous survey on eating habits and food-intake frequency in a sample of college students [89]. The regression analysis at T0 and T1 found the same risk factors as the total sample except for the variable named living alone, which does not appear to be a risk factor at T1. As expected, many students went back to live with their parents during the pandemic, which could explain the difference with T0.

### Strengths and Limitations

University students are a fascinating group to study about lifestyle and health behaviours. Investigating university students’ behaviours can be very useful in order to guide universities in setting up specific prevention strategies and awareness campaigns.

Our study was one of the first to investigate physical activity and eating behaviour in university students with data collection both before and during the COVID-19 pandemic.

However, the study has several limitations that should be addressed:

The sample lacks statistical representativeness due to the sampling procedure and the non-probabilistic strategies that could have led to a selection bias. However, using social media to promote the survey can be considered a low cost and fast way to collect data, especially in a pandemic situation [90].

A cross-sectional design could also be considered a limit since it measures the cause and the effect at the same point in time and cannot support findings on causal relationships. This design represents a clear and standardised method to determine the prevalence of health-related behaviour, such as wearing seat belts or participating in exercise [91,92].

Another limitation of the study is the validity of self-reported measures that could lead to overestimating or underestimating physical activity behaviours and being prone to inaccuracies when collecting data on eating behaviours [49,93,94,95]. Future studies should use more objective measures to perform a more precise assessment of these parameters [70].

Although the instrument used showed good statistical power, the questions concerning PA did not allow a very precise assessment of the energy expenditure. The use of a validated scale such as the International Physical Activity Questionnaire (IPAQ) [96], for example, would have allowed a better calculation of the MET expenditure and more accurate identification of PA levels. An assessment of the regulatory styles (intrinsic/extrinsic) in the students’ intention to practise PA could also have helped provide further useful insights into the comparison between before and during the pandemic lockdown [97].

Moreover, when investigating eating habits, the specific dietary components intake and healthy dietary patterns were not assessed. Future research should try to use validated scales to build a composite diet quality index such as the Diet Quality Index - International [98] and the Healthy Eating Index [99] to make the statistical analysis and the discussion of the results easier. However, previous studies already used meal frequency to assess the association with the prevalence of obesity and cardiovascular diseases [61].

The questions used to investigate physical activity and eating habits are adapted from the HBSC questionnaire. The items of the HBSC physical activity questionnaire have acceptable reliability and validity among international students [51].

When the first data collection was performed, our aim was to investigate several lifestyle habits at the same time in school-aged students and university students. Therefore, we decided to use questions from a validated international tool (translated into Italian) for its comparability value. When the second collection took place, even if the objective was to focus only on two habits, we decided to use the same questions to make the comparison more accessible and more statistically valid.

In addition, we acknowledge that the second version of the questionnaire lacks questions related to the exposure to COVID-19 infection (directly or indirectly). This variable could have influenced the behaviours associated with PA and a healthy diet.

Another possible limitation of the study could be the presence of selection biases and residual confounding bias. We tried to limit these biases with a sampling method (at T0 and T1) based on simple randomisation, a good size of the sample, and a regression analysis that could support us on the validity of the data that emerged.

Finally, one of the primary limits of the study was that participants did not coincide at T0 and T1 since when we collected the first data (T0), our aim was not to have a longitudinal investigation. This limit is common to other studies published during the unexpected event of the COVID-19 pandemic with different populations [37,49,100]. In order to reduce this limitation, using the same inclusion criteria is necessary to obtain two similar subsamples. In our case, we also attempted to collect data during the same time of year, which allowed a comparison in similar conditions of university commitment and general availability of leisure time (classes period instead of exams period).

## 5. Conclusions

Our findings suggested that during home isolation due to the COVID-19 pandemic, PA and eating behaviours were negatively affected in university students.

During the pandemic, the tendency of young people to gain weight, spend too many hours without moving, and adhere to healthy eating seems to have worsened, with differences between genders: male students seem to be more active than females but less careful about eating behaviours.

Adverse health behaviours tend to increase with age and become more pronounced when young people acquire autonomy and can therefore express their own preferences.

Nutrition and physical activity were central themes in health campaigns in the early nineties, but this centrality faded in the following years. Only recently, the topic has been taken up again, and a more significant information effort towards the whole population is needed since improving healthy eating behaviours and increasing PA levels are both social and individual responsibilities. Therefore, a multi-sector, multidisciplinary and culturally relevant population-based approach is required.

In an emergency situation such as the COVID-19 pandemic, it seems that the relationship among health behaviours is even more vital. Previous studies suggest that physical exercise leads to healthier nutritional choices, and psychological states, in turn, influence the decisions of university students [46]. Psychological pathways, especially motivation, behavioural intentions, and anxiety, could influence the adherence to healthy behaviour and should be further investigated in future studies in order to develop the best strategy to put in place, especially during emergency periods such as a pandemic [101].

Universities, not only the ones focussing on scientific and medical areas, should invest in the protection and promotion of health of their students with specific awareness programmes and include the protection and promotion of health in their core values (8).

Further research should repeat the survey in the post-lockdown period to investigate the long-term effects on physical activity, sedentary behaviour, and nutrition behaviours.

## Figures and Tables

**Table 1 ijerph-19-05550-t001:** Characteristics of the study population: total sample, sample at T0 and sample at T1.

	Total (*n*. 2001)	T0 (*n*. 1025)	T1 (*n*. 976)
*Age* (*average)*	22.7 years ± 5.5 SD	22.6 years ± 3.6 SD	21.3 years ± 4.1 SD
Gender (%)			
male	39.8	35.5	31.3
female	60.2	64.5	68.7
*Fathter Educational level (%)*			
no formal education	0.8	0.9	0.6
primary	19.0	16.8	19.9
secondary	41.3	42.6	40.6
university	38.9	39.7	39.9
*Mother Educational level (%)*			
no formal education	1.1	1.3	0.8
primary	15.2	16.2	15.5
secondary	41.3	39.4	40.0
university	42.4	43.1	43.7
*Relationship Status (%)*			
live with their family	59.2	54.2	82.9
live alone	29.7	36.4	12.8
other	11.1	9.4	4.3
*Area of study (%)*			
Scientific	16.9	18.1	17.1
Humanities	44.2	43.0	42.2
Health sciences	29.9	31.4	32.3
Legal/Business	9.0	7.9	8.4

The values of Cronbach’s alpha (coefficient of internal consistency) for the questionnaire used in the study was 0.74 and 0.78, respectively, for the study carried out at T0 and T1.

**Table 2 ijerph-19-05550-t002:** Statistically significant changes of physical activity, eating behaviours, BMI, and perception of own body weight by gender.

	Total (*n*. 2001)		T0 (*n*. 1025)		T1 (*n*. 976)	
	Male	Female	Male	Female	Male	Female
*Regular PA practice (%)*						
Yes (1–2 times per week)	53.1	46.9	56.9	43.3	60.8	39.2
*PA practice motivations (%)*						
Lose weight	22.2	67.6	25.7	56.9	19.9	61.1
*Daily meals frequency (%)*						
5	11.1	15.7	10.3	14.4	31.3	40.0
3	59.9	40.4	58.7	47.8	59.0	57.2
2	30	29.2	31.0	32.3	7.0	5.9
*Eating Breakfast (%)*						
Rarely	59.8	39.7	50.3	33.7	62.8	42.9
*Have been on a diet* *(%)*						
at least once	8.2	76.9	9.1	74.4	7.7	71.1
*BMI* *(%)*						
Overweight	21.6	17.9	28.5	20.1	33.5	29.2
Obese	8.9	9.8	6.7	10.1	9.9	10.7
*Perception of own weight status* *(%)*						
Overweight	32.3	72.1	29.4	68.2	23.1	77.8

Statistically significant differences *p* < 0.001.

**Table 3 ijerph-19-05550-t003:** Logistic regression models relating some sociodemographic variables and PA in the total sample and the two subsamples (T0 and T1).

		Physical Actvity	
		(PA)	
		OR	95% CI
**TOTAL SAMPLE (** ** *n* ** **. 2001)**			
*Gender*	Female	1	
	Male	1.49	1.12–2.58
*Age*	≤25	1	
	≥26	1.52	1.29–2.72
*Living with parents with a low level of education*	No	1	
	Yes	1.36	1.15–1.88
*Have a positive body perception*	No	1	
	Yes	4.2	2.10–6.90
**T0 (*n*. 1025)**			
*Gender*	Female	1	
	Male	1.31	1.17–1.97
*Age*	≤25	1	
	≥26	1.39	1.08–1.78
*Living with parents with a low level of education*	No	1	
	Yes	1.85	1.05–3.25
*Have a positive body perception*	No	1	
	Yes	2.61	1.91–4.98
**T1 (*n*. 976)**			
*Gender*	Female	1	
	Male	1.61	1.32–1.96
*Age*	≤25	1	
	≥26	1.36	1.04–2.71
*Living with parents with a low level of education*	No	1	
	Yes	1.57	1.07–3.48
*Have a positive body perception*	No	1	
	Yes	2.18	1.89–3.51

Statistically significant differences *p* < 0. 001.

**Table 4 ijerph-19-05550-t004:** Logistic regression models relating some sociodemographic variables and eating habits in the total sample and the two subsamples (T0 and T1).

		Eating Habits	
		OR	95% CI
**TOTAL SAMPLE (n. 2001)**			
*Gender*	female	2.7	2.21–4.90
	male	1	
*Age*	≤25	2.41	1.91–3.22
	≥26	1	
*Relationship Status*	live alone	1.45	1.06–2.91
	live with family	1	
**T0 (n. 1025)**			
*Gender*	female	2.85	1.77–4.25
	male	1	
*Age*	**≤25**	3.09	2.41–6.59
	**≥26**	1	
*Relationship Status*	live alone	1.29	1.01–2.21
	live with family	1	
**T1 (n. 976)**			
*Gender*	female	3.81	2.22–4.59
	male	1	
*Age*	≤25	3.57	2.55–5.00
	≥26	1	
*Relationship Status*	live alone		---
	live with family		---

Statistically significant differences *p* < 0.001.

## Data Availability

The data presented in this study are available on request from the corresponding author.

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
