# Peer review of "Changes in Physical Activity Levels and Eating Behaviours during the COVID-19 Pandemic: Sociodemographic Analysis in University Students"

_ijerph, 2022, doi:10.3390/ijerph19095550_

Round 1

Reviewer 1 Report

  • Excellent paper
  • Well organize and presented.
  • Actual and high references.
  • It would be a good idea to detail a little bit more how and the procedure about the comparison in similar conditions of university commitment as suggested by other studies. 
  • In the design and study subjects it would be desirable to detail the age and gender percentages of the participants.

Author Response

Dear reviewer,

Thank you very much for your comments.

As per your suggestion, age and gender characteristics of the two samples (that were only indicated in table 1) have been added in the methods section of the paper.

The comparison between the two samples has been possible considering that, as other previous studies during the pandemic, we used the same inclusion criteria and we chose to collect data from the two samples during the same months of the year. This means that, in both data collection, the students were busy with classes rather than exams (different availability of leisure time). In order to clarify this concept, some details have been added to the text and a reference has been deleted because we realized was not entirely proper to explain this concept.

Reviewer 2 Report

The authors have put together a paper looking cross sectionally at changes in PA, Diet, and Weight amongst Italian college students at two separate time points. This research provides an interesting look at how habits changed during COVID, which is a unique natural experiment we have all been living through.

Generally, the paper is well written, however, I would ask that all references to ‘girls’ be changed to women, and the same for the few references to ‘boys’, which should be men.  One other are of writing that could be reconsidered is the frequent use of one sentence paragraphs, such as appear on lines 65-76. Please consider merging these as needed. Also, see some obvious typos on line 263.

In the Methods, I have a few questions.

-Line 171 indicates two categories were created (sedentary/active), however, I was unable to determine the specifics of how students were put into either category. Was it based on one response item, and if so, which?

-Line 171-173 states that for ‘busy’ students, other categories were made, but it is not clear how ‘busy’ was defined. Also, the three categories include ‘low frequency’, which is defined as “less than one time per week”. Help me understand how ‘less than one time per week’ is defined as ‘low frequency’ instead of sedentary. Further, what are the three categories based on, as I would argue that 1 weekly bout of exercise should not be defined as ‘medium frequency’. In fact, by most standards, it is a dichotomous standard, you either are or are not meeting the threshold for active. Therefore, please explain this rationale.

-Line 231 appears to be a typo in the meal frequency, as it lists meals per day category as (<5 or 5). Further, in the results, the meal frequency categories are 5, 3, 2; are those the only options respondents were given? It would seem this should be an open-ended question.

-Table 1: Characteristics, i believe has a typo under Area of Study. Based on your text, the second category should read Humanities, not Human Sciences?

-My only other inquiry is around BMI reporting. Given that you used self-report H/W, which has its own inherent bias and lacks the sensitivity of actual measures, does it really make sense to report changes over such a short period of time. My point being, will self-reported height and weight change much in a 1 year window? In addition, you indicated it is not the same cohort, which to me makes the BMI comparison less valid. (is it possible to identify those who repeated the survey and assess them separately?).

Author Response

Dear reviewer,

thank you very much for your comments. 

As per your suggestion, we change the words girls/boys to women/men, we checked the typing mistakes you pointed out and we tried to merge the sentences related to the same subtopic in the text.

We copied your comments and answered in red color.

In the Methods, I have a few questions.

-Line 171 indicates two categories were created (sedentary/active), however, I was unable to determine the specifics of how students were put into either category. Was it based on one response item, and if so, which?

The students were asked if they used to perform any physical activity. The possible answers were yes/no. More detailed questions about PA and sport were proposed only to students answering “yes”. We tried to better specify it in the text.

-Line 171-173 states that for ‘busy’ students, other categories were made, but it is not clear how ‘busy’ was defined. Also, the three categories include ‘low frequency’, which is defined as “less than one time per week”. Help me understand how ‘less than one time per week’ is defined as ‘low frequency’ instead of sedentary. Further, what are the three categories based on, as I would argue that 1 weekly bout of exercise should not be defined as ‘medium frequency’. In fact, by most standards, it is a dichotomous standard, you either are or are not meeting the threshold for active. Therefore, please explain this rationale.

Unfortunately the questions in the questionnaire didn’t allow us to create precise categories considering the thresholds of the WHO recommendations since the options for students that declared to do some PA were: 1 time a month; less than 1 time per week; 1-2 times per week and 3 or more times per week and we didn’t have information on the total amount of minutes per week. In the new version we changed the labels of the categories in order to avoid misunderstanding.

We changed the name of the categories as follows: very low, low and medium-high. In this way, it should be easier to imagine and hypothesize that students with very low and low frequency don’t reach the threshold.

-Line 231 appears to be a typo in the meal frequency, as it lists meals per day category as (<5 or 5). Further, in the results, the meal frequency categories are 5, 3, 2; are those the only options respondents were given? It would seem this should be an open-ended question.

In methods section there was a typo mistake that we corrected. For every type of meal (breakfast, morning snack, lunch etc..) the students indicated if they use to have it or no. The sum of “yes” gave the number of meals per day.

In table 2  (where you found 5,3,2) we only reported the statistically significant data as the number of students who reported having 1 or 4 meals a day is numerically negligible and not statistically significant.

-Table 1: Characteristics, i believe has a typo under Area of Study. Based on your text, the second category should read Humanities, not Human Sciences?

We changed it.

-My only other inquiry is around BMI reporting. Given that you used self-report H/W, which has its own inherent bias and lacks the sensitivity of actual measures, does it really make sense to report changes over such a short period of time. My point being, will self-reported height and weight change much in a 1 year window? In addition, you indicated it is not the same cohort, which to me makes the BMI comparison less valid. (is it possible to identify those who repeated the survey and assess them separately?).

The comment is interesting and the objection is probably valid for short-term analysis during a period different than the COVID-19 pandemic.

As per Ref. 86 (Pop C, Ciomag V. Impact of COVID-19 lockdown on body mass index in young adults. Physical education of students. 2021;25(2):98-02) it seems that, also in another a study on University students, during the pandemic BMI increased with 1.8 kg/m2, primarily because of weight gain. Since our two samples are very similar, we decided to show the difference at T0 and T1. The readers should consider this data as an estimate since the data are also self-reported.

Reviewer 3 Report

This is an interesting manuscript dealing with a timely topic, although many methodological points remain still unsolved.

Abstract

-The authors referred to a sampling strategy similarly to a previous one, but this still unclear.

-More details on the sample are required (e.g., age)

Introduction

-Overall, it is well-conducted although the authors should put more focus on the university students´ populations.

Methods

-The way you conducted your sampling is called at convenience.

-Do you have any idea on how many participants refuse to participate?

-A flowchart depicting the number of participants for each wave would be useful.

-Were the questions assessing health behaviours validated?

-This is my main concern, since the logistic regression model seems univariate and confusion might not have been controlled, thus, a confusion bias possibly explain your results.

Results

-Check “Avarage” in Table 1

-In Table 3 and Table 4, authors should state which variables did they use to adjust the model.

-It is curious that ORs are your main results but you don´t provide them in the abstract.

Discussion

-The first paragraph should summarize your main results and comment on what this adds to the state of the art of the topic.

-Lines 438-429. I have doubts that this is the first study investigating university students eating habits during the pandemic.

-Selection bias, and, importantly, residual confounding bias, might distort the reality in a way that your estimates are far from the truth.

Conclusion

-You should be more humble, since it is not possible to demonstrate what you are stating in this paragraph.

Author Response

Dear reviewer,

thank you for the time dedicated to read and comment our paper. We appreciate your suggestions and concerns and we have tried to address your comments in the new version of the paper (in red color).

We copied your comments and replied in red color.

Abstract

-The authors referred to a sampling strategy similarly to a previous one, but this still unclear.

We added to the abstract the sampling approaches used.

-More details on the sample are required (e.g., age)

We added information about gender and age

Introduction

-Overall, it is well-conducted although the authors should put more focus on the university students´ populations.

As suggested, we added two phrases on the University students population in order to better focus the study.

Methods

-The way you conducted your sampling is called at convenience.

We added in the text

-Do you have any idea on how many participants refuse to participate?

We have the data on refusal to participate for the first data collection, which took place in paper form (77 refusals out of 1102), but not for the second collection, which, as explained in the methodology, took place online. So we have decided, in order to standardise the information, to omit the data in the paper.

-A flowchart depicting the number of participants for each wave would be useful.

Thank you for the suggestion. As explained before, methodologically it did not seem right to do this only on the sample at T0 for which we had the information.

-Were the questions assessing health behaviours validated?

Yes, as we mentioned in the methods section, we used the questions from the Health behaviour in School Children (HBSC) survey since the questionnaire was initially submitted to both, school-aged students and university students.

-This is my main concern, since the logistic regression model seems univariate and confusion might not have been controlled, thus, a confusion bias possibly explain your results.

The regression models were built after the results of the bivariate analysis precisely to try to limit and control confounding bias as much as possible.

Results

-Check “Avarage” in Table 1

Done, corrected to Average

-In Table 3 and Table 4, authors should state which variables did they use to adjust the model.

We added a paragraph in the methods in the statistical analysis section before the explanation of the logistic model. The adjustment for sociodemographic characteristics (age, gender, area of study, parents' level of education, perception of own body) took place through the coding of the sociodemographic variables that could influence the behaviour of our sample and consequently some dummy variables were created, and the possible effects of the changes on the dependent variables (PA and Eating Habits) were evaluated.

-It is curious that ORs are your main results but you don´t provide them in the abstract.

Due to the word limits in the abstract we decided not to write the ORs numbers but we added a short phrase to better explain the results.

Discussion

-The first paragraph should summarize your main results and comment on what this adds to the state of the art of the topic.

We added a paragraph with the main findings at the beginning of the Discussion and we compared our results with other studies in the following paragraphs.

-Lines 438-429. I have doubts that this is the first study investigating university students eating habits during the pandemic.

We have modified the phrase.

-Selection bias, and, importantly, residual confounding bias, might distort the reality in a way that your estimates are far from the truth.

We added in the Limits paragraph: “A possible limitation of the study could be the presence of selection bias and residual confounding bias. We tried to limit this bias with a sampling method (at T0 and T1) based on simple randomisation, a good size of the sample and a regression analysis that could support us on the validity of the data that emerged”

Conclusion

-You should be more humble, since it is not possible to demonstrate what you are stating in this paragraph.

We agree on the suggestion therefore we changed some words in the text and added some phrase in order to address this comment.

Round 2

Reviewer 3 Report

The authors have subtantially improved the quality of the manuscript.